# Systematic Review of the Psychopathological Symptomatology and Neuropsychological Disorders of Chronic Primary Musculoskeletal Pain

**DOI:** 10.3390/healthcare12151465

**Published:** 2024-07-23

**Authors:** Alejandro Arévalo-Martínez, Carlos Barbosa-Torres, Juan Manuel Moreno-Manso, Pilar Cantillo-Cordero, María Elena García-Baamonde, César Luis Díaz-Muñoz

**Affiliations:** 1Department of Psychology, Faculty of Education and Psychology, University of Extremadura, 06071 Badajoz, Spain; aarevaloj@unex.es (A.A.-M.); jmmanso@unex.es (J.M.M.-M.); pcantillo@unex.es (P.C.-C.); mgarsan@unex.es (M.E.G.-B.); 2Department of Medical-Surgical Therapy, Faculty of Medicine and Health Sciences, University of Extremadura, 06071 Badajoz, Spain; cdiazmun@unex.es

**Keywords:** psychopathology, neuropsychology, systematic review, chronic pain, chronic primary musculoskeletal pain

## Abstract

Chronic pain can develop without tissue damage, injury, or underlying illness. There are several intervening biological, psychological, and social factors involved in its appearance that significantly affect the activities of daily life. It is also associated with significant emotional anxiety and/or functional disability. This review systematically analyses works published in the last five years that evaluate the psychopathological symptomatology and neuropsychological disorders of chronic primary musculoskeletal pain (CPMP). A bibliographic search was carried out to identify articles published in English between January 2018 and March 2023 using the Medline, Scopus, PsycInfo, and Pubmed databases. Twenty articles were obtained using the PRISMA selection method. The main results of this study provided evidence of the presence of moderate and severe chronic pain in patients suffering from musculoskeletal pain. This increase in the intensity of pain correlates with greater psychopathological symptomatology, such as depression, anxiety, insomnia, lack of attention, and hyperactivity/impulsiveness, as well as the use of maladaptive coping strategies. Furthermore, there exists dysfunction in the cerebral structures related to attention and the processing of pain in patients with CPMP. This review may help to develop and optimise the multidisciplinary treatments adapted to the deficits caused by this illness.

## 1. Introduction

Chronic pain is one of the main causes of needing medical attention because of the important repercussions for the physical and psychological health of those who suffer from it [1]. Despite the fact that the concept of chronic pain has historically been associated with the existence of some kind of organic damage, the appearance of biopsychosocial explanations has meant that there is a new way to understand it, mainly in cases in which there is no identifiable organic pathology [2,3].

According to the biopsychosocial model, the incapacitating nature of chronic pain is the result of a multidimensional interaction among biological, psychological, and social factors [2,3,4]. Generally speaking, depression, anxiety, distress, sleep problems, somatic symptoms, maladaptive coping strategies, and pain catastrophizing have been considered psychopathological disorders caused by chronic pain. However, some works of research have demonstrated that these disorders may also act as risk factors for the development of this pathology [1,3,4,5]. On a neuropsychological level, chronic pain has been associated with a slight deterioration in cognitive functions, in particular, memory, attention, and executive functions [6,7]. Similarly, some recent reviews have demonstrated the existence of dysfunctions in the brain’s structures in patients suffering from chronic pain. The most affected areas are related to nociception, the most important being the somatosensory cortex, the insular cortex, the anterior cingulate cortex, and the prefrontal cortices [3,6,7]. In addition, studies also show an increase in the resting state functional connectivity (rsFC) in patients with chronic pain; this may be related to anxiety, depression, pain catastrophizing, and deficits on a cognitive level [7].

In the last revision of the International Classification of Diseases (ICD-11), the World Health Organization (WHO) established a new definition and classification for chronic pain to eliminate the deficiencies identified in the previous versions [8], defining it as an “unpleasant sensory and emotional experience associated with, or resembling that associated with, actual or potential tissue damage. Chronic pain is pain that persists or recurs for longer than 3 months” [9]. As an additional novelty to this conceptualisation, primary and secondary chronic pain are differentiated. For chronic primary pain, the symptoms have a multifactorial origin and are not the result of an underlying illness, while for chronic secondary pain, the symptoms are the result of an underlying illness [8,10]. Figure 1 shows the classification of chronic pain and its respective diagnostic characteristics.

Epidemiological studies conducted globally have shown a significant increase in diagnoses of chronic pain in recent decades [11,12,13]. Specifically, the most frequently reported types of chronic pain are musculoskeletal in origin, often localized in areas such as the back, shoulders, neck, hips, knees, wrists, and feet [1,14,15]. Additionally, alongside physical and psychological distress, individuals with chronic musculoskeletal pain also experience difficulties in engaging in social activities and performing their work tasks effectively [16,17]. 

Because of its high prevalence compared with other chronic pain conditions and the significant impact the disease has on the lives of those who experience it, this systematic review will focus on chronic primary musculoskeletal pain (CPMP) [1,18]. Within the framework of the ICD-11, CPMP is presented as a new diagnostic category when the aetiology is not clear but biological, psychological, and social factors that significantly affect the daily activities of life are all involved in its appearance [19,20]. The ICD-11 establishes the following criteria for its diagnosis [9]:(A)Chronic primary pain.(B)Located in the muscles, bones, joints, or tendons.(C)Associated with significant emotional distress and/or functional disability.(D)The diagnosis is applicable independently of the biological or psychological factors, except when another diagnosis explains the symptoms more exactly.

The following subtypes are included in this category: chronic primary low back pain, chronic primary cervical pain, chronic primary thoracic pain, and chronic primary limb pain [9]. Although research aiming to deepen the understanding of CPMP is limited, Fitzcharles et al. [19] proposed several distinctive features of CPMP compared with chronic secondary musculoskeletal pain. In this regard, CPMP is predominantly nociceptive and described as a sharp, stabbing, or burning sensation [8,21]. Individuals with this diagnosis often exhibit pain hypersensitivity, as well as a variable and diffuse pain experience that may radiate to other areas of the body [22,23]. Moreover, increased pain intensity is common and correlates with experienced psychosocial stress [4,10]. CPMP has also been associated with a higher prevalence of psychopathologies, cognitive impairment, and other comorbid conditions [4,19]. On the other hand, chronic secondary musculoskeletal pain is predominantly nociceptive [10,18]. This type of pain is described as a localized pressure sensation near the injury, with infrequent hypersensitivity and stable pain intensity [24,25]. Although quality of life is generally impaired in both diagnoses, CPMP may lead to greater deterioration because of its high prevalence of psychopathological and physical comorbidities [4,18,26]. 

Despite the considerable increase in research dealing with the identification of the existing disorders at a psychological level in patients suffering from chronic pain, the recent conceptualisation from the World Health Organization (WHO) concerning this pathology can suppose a starting point for a better understanding of the diagnosis and the consequences associated with each one of the newly proposed diagnostic categories [20]. In line with the conclusions raised by Treede et al. [10], we are currently at the right moment to enhance the understanding of chronic primary pain and develop multimodal treatments tailored to the new diagnostic classifications. Therefore, to intervene comprehensively in the population with CPMP, it is essential to assess their needs thoroughly beforehand. Thus, the objective of the current review is to analyse systematically the research works published in the last five years that evaluate the psychopathological symptomatology and the neuropsychological dysfunctions of CPMP. In order to guarantee homogeneous and accurate results with respect to future research, the inclusion criteria established by the World Health Organization (WHO) in the ICD-11 are used here.

## 2. Materials and Methods

This study was conducted following the Preferred Reporting Items for Systematic Reviews and Meta-Analyses (PRISMA) guidelines [27] (Appendix A Appendix A: PRISMA checklist).

### 2.1. Selection Criteria

To be included in this review, the publications had to fulfil the following additional criteria: (a) to be an empirical research work that evaluates the psychopathological symptomatology and/or the neuropsychological dysfunctions in any of the subtypes of CPMP; (b) to use a sample of adults between 18 and 95 years of age; (c) the CPMP should not be the consequence of an injury or illness; (d) the sample should not have additional diseases, such as cardiovascular, respiratory, metabolic, or neurological conditions; (e) to use psychopathological and/or neuropsychological evaluation tests; (f) to include the results of the psychopathological and/or neuropsychological evaluation in the baseline; and (g) to include the necessary data to determine the existence of psychopathological symptomatology and/or neuropsychological dysfunctions in the sample.

The following were the exclusion criteria: (a) publications in which the abstract makes no reference to CPMP or any of its subtypes, or to the related psychopathological and/or neuropsychological variables; (b) publications that do not specify the subtype of CPMP or, when specified, do not form part of the objectives of the publication; and (c) publications that do not distinguish between CPMP and other types of chronic primary pain, such as chronic widespread pain, including fibromyalgia syndrome.

### 2.2. Search Strategy

A bibliographic search was carried out to identify papers published in English in journals over the last five years (between January 2018 and March 2023) using the Medline, Scopus, PsycInfo and Pubmed databases. Because of the existence of different diagnostic subtypes in CPMP and its relation with the dysfunctions being studied here, a combined search was carried out using the following terms: (chronic primary musculoskeletal pain OR chronic musculoskeletal pain OR chronic primary low back pain OR chronic primary cervical pain OR chronic primary thoracic pain OR chronic primary limb pain) AND (psychopathology OR mental disorder OR mental illness OR neuropsychology OR neuropsychological test OR neuropsychological assessment OR cognitive impairment OR cognitive dysfunction OR cognitively impaired OR executive function OR cognitive function OR cognitive performance OR memory OR attention).

A.A.-M., C.B.-T., and J.M.M.-M. worked together to select the keywords and search criteria. A.A.-M. individually evaluated the title and abstract of each article according to the established inclusion and exclusion criteria. Subsequently, A.A.-M., C.B.-T., and J.M.M.-M. independently examined the full text of each potentially eligible article. Disagreements were resolved through discussion among these reviewers and, if necessary, the opinion of P.C.-C. was sought.

### 2.3. Quality Assessment and Risk of Bias

The methodological quality of the studies was independently assessed in pairs (A.A.-M./C.B.-T. and J.M.M.-M./M.E.G.-B.) using the Newcastle–Ottawa Scale (NOS) [28]. The NOS scale evaluates the quality of studies based on the dimensions of selection of study groups, comparability of the groups, and outcomes/exposure, making it the most suitable tool for the studies included in this review. Any disagreement among the reviewers was discussed with P.C.-C. to avoid the risk of biases. Finally, there was consensus in the selection of the studies. The assessment of methodological quality was overseen by C.L.D.-M.

### 2.4. Data Extraction and Analysis

The reviewers, organised in pairs (A.A.-M./C.B.-T. and J.M.M.-M./M.E.G.-B.), independently extracted and analysed data from the eligible studies. A series of variables was gathered concerning the studies reviewed related to their design and methodology. These included (a) the country where the research was performed, (b) the study design, (c) the subtypes of CPMP, (d) the existence of a control group and/or additional groups, (e) the number of participants in the study, (f) the age and gender of the participants, (g) the instruments used to evaluate CPMP, psychopathological symptomatology, and neuropsychological dysfunctions, (h) the variables related to CPMP, the psychopathological symptomatology, and the neuropsychological dysfunctions, (i) the results of the evaluation of CPMP, psychopathological symptomatology, and neuropsychological dysfunctions. In the case of disagreement among the reviewers, P.C.-C. was consulted. The extraction process was overseen by C.L.D.-M.

## 3. Results

### 3.1. Selection of the Studies and Their Characteristics

Starting from the database search, a total of 3578 articles were obtained. First, 282 duplicated articles and 2424 articles not related to the question at hand were eliminated. Then, 872 articles were reviewed with respect to their title and abstract, with 816 articles being excluded as they did not comply with the objective of this review. In total, 56 articles were identified concerning psychopathological symptomatology and neuropsychological dysfunctions. However, after reading the complete texts, another 36 articles were excluded as they did not comply with the established inclusion criteria. Finally, a total of 20 articles [29,30,31,32,33,34,35,36,37,38,39,40,41,42,43,44,45,46,47,48] were included in the review. The details are shown in the PRISMA flow diagram [27] in Figure 2.

The research works included in the current review have wide geographical heterogeneity, coming mainly from the USA (30%) [29,37,43,44,45,48], Japan (15%) [34,36,47], Germany [30,40], Australia [33,42], and Norway [35,46] (10% each). Other articles originated from Belgium [32], Brazil [39], China [38], Iran [41], and New Zealand [31] (5% each).

### 3.2. Study Design, Subtypes of CPMP, and Existence of a Control Group and/or Additional Groups

Table 1 summarises the main characteristics of the articles included in this review. The information concerning the other variables, as well as the psychopathological symptomatology and neuropsychological dysfunctions, are shown in Table 2 and Table 3, respectively. The majority of the research works used a cross-sectional study (80%) [29,30,31,32,33,34,35,36,37,38,39,42,43,45,47,48], while a smaller percentage used a longitudinal study (20%) [40,41,44,46].

In 17 of the articles (85%) [29,30,31,32,33,34,36,37,38,39,40,41,42,44,45,46,48], the research was carried out in a healthcare context and the diagnoses were based on the participants’ medical records. In the reviewed studies, 16.4% of the participants were diagnosed with CPMP, while the remaining 83.6% were in the control group with no type of diagnosis or diagnosis related to this pathology. As for the subtypes of CPMP analysed, chronic primary low back pain (CPLBP) was the most numerous diagnosis among the participants included in this review, being present in 18 of the articles (90%) [30,31,33,34,35,36,37,38,39,40,41,42,43,44,45,46,47,48], followed by chronic primary cervical pain (CPCP) in 2 of the articles (10%) [29,32].

As for the presence of a control group in the reviewed studies, the majority (55%) [30,31,33,34,36,40,41,43,46,47,48] used just one control group that included individuals with the pathology being studied, while the rest used a healthy control group with no type of diagnosis (45%) [29,32,35,37,38,39,42,44,45]. Of the studies that used one control group, only Coppieters et al. [32] included a group with a different pathology to compare with CPCP (5%), this group having a diagnosis of neck pain as a consequence of physical trauma.

### 3.3. Number, Gender, and Age of the Participants

The reviewed studies evaluated a total of 42,702 participants, the smallest sample size being 13 [47] and the largest 30,669 [35]. The total number of women (55%) was slightly higher than that of men (45%). Furthermore, the participants’ average age was 45.7 years (SD = 12.57), and the age range oscillated between 18 and 95.

In comparison with the total number of participants, the proportion of women with CPMP (61%) was considerably larger than that of men with the same diagnosis (39%), while the average age in years was 46.6 (SD = 11.98).

### 3.4. Instruments of Evaluation and Symptoms and/or Psychopathological and Neuropsychological Dysfunctions of CPMP

The information concerning the instruments used to evaluate the subtypes of CPMP, psychopathological symptomatology, and neuropsychological dysfunctions are shown in Table 2 and Table 3. The instruments used in the reviewed studies varied widely, with a total of 63 different instruments used to carry out these evaluations. The Numeric Rating Scale (NRS) (7%), the Pain Catastrophizing Scale (PCS) (5%), the Beck Depression Inventory (BDI) (4%), functional magnetic resonance imaging (fMRI) (4%), and the Roland–Morris Disability Questionnaire (RMDQ) (3%) were the most commonly used instruments. Using these instruments, a total of 57 symptoms and/or dysfunctions were evaluated, with depression (10%), intensity of the pain (9%), anxiety (7%), somatic symptoms (5%), and brain activity (4%) being the most commonly evaluated.

Of the 63 instruments used in the reviewed studies, 26 (41%) were aimed at evaluating CPMP and its subtypes. The most commonly used were the Numeric Rating Scale (NRS) (15%), the Pain Catastrophizing Scale (PCS) (11%), the Roland–Morris Disability Questionnaire (RMDQ) (7%), the Visual Analogue Scale (VAS) (7%), and the STarT back screening tool (SBT) (4%). These instruments evaluated a total of 22 different symptoms and/or dysfunctions (39%) with the intensity of the pain (22%), pain catastrophizing (10%), the incapacity caused by the pain (10%), the impact of the pain on daily activities (8%), and the general state of health (6%) being the most commonly evaluated.

Regarding psychopathological symptomatology, a total of 27 instruments (43%) were aimed at their evaluation, with the Beck Depression Inventory (BDI) (12%), the Center for Epidemiological Studies-Depression (CES-D) (6%), the Fear-Avoidance Beliefs Questionnaire (FABQ) (6%), the Hospital Anxiety and Depression Scale (HADS) (6%), and the Patient Health Questionnaire Depression Scale (PHQ-2/PHQ-9) (6%) being the most commonly used. These instruments evaluated a total of 27 different symptoms (47%), of which depression (23%), anxiety (17%), somatic symptoms (11%), sleep problems (8%), and maladaptive coping strategies (6%) were the most commonly evaluated.

With respect to neuropsychological dysfunctions, a total of 10 instruments (16%) were used for their evaluation, with functional magnetic resonance imaging (fMRI) (29%), the electroencephalogram (EEG) (14%), a battery of seven cognitive tests (7% each), and Cogstate’s digital cognitive evaluation (7%) being the most used. Using these instruments, a total of eight different dysfunctions (14%) were evaluated, with brain activity (29%), attention (19%), cerebral images (19%), executive functions (10%), and memory (10%) being the most commonly evaluated.

### 3.5. Results of the Evaluation of CPMP, Psychopathological Symptomatology, and Neuropsychological Dysfunctions

#### 3.5.1. Chronic Primary Musculoskeletal Pain

The severity of CPMP and its subtypes were evaluated in a total of 12 articles (60%) [29,30,32,33,36,37,39,40,41,42,43,44]. The majority of the participants presented a mean intensity of chronic pain in muscles, bones, joints, and/or tendons between moderate and severe. Worth noting is the study by Neumann and Hampel [40], which classified the sample with chronic primary low back pain (CPLBP) into three progressive stages of pain. Among those who showed evidence of high levels, significant differences were found in the duration of pain, its location, its mean intensity, and deterioration in physical health. These differences were greater as the pain became chronic. The analysis of the existing correlations between the intensity of pain and the other symptoms studied in four of these articles (33%) [30,33,43,44] showed evidence of positive correlations with pain catastrophizing, pain management, depression, anxiety, sleep problems, symptoms arising from a lack of attention, and hyperactivity/impulsiveness and coping strategies, while negative correlations were found with beliefs concerning pain control, the acceptance of pain, and the quality of sleep. Pain catastrophizing was analysed in four of the articles (33%) [33,37,39,42], and the patients with CPMP showed significantly higher levels than those with no pathology. The analysis of the existing correlations between pain catastrophizing and other symptoms showed evidence of positive correlations with depression, anxiety, avoidance behaviour and neuroticism, while negative correlations were found with pain management, the acceptance of pain, and happiness. 

Similarly, in patients suffering from CPMP, the studies analysed showed evidence of significant levels of kinesiophobia and fear of physical activity, in addition to a pain threshold significantly lower than the participants with no pathology. The use of opioids was analysed in two of the articles (17%) [42,43], and the population with CPMP showed a strong tendency towards abusive consumption.

#### 3.5.2. Psychopathological Symptomatology

The psychopathological symptomatology of CPMP and its subtypes were evaluated in 17 of the articles (85%) [30,31,33,34,35,36,37,38,39,40,41,42,43,44,45,46,48]. Symptoms of depression (present in 71% of these articles) [30,33,34,35,37,39,40,42,43,44,45,48], anxiety (present in 47%) [30,31,33,34,35,39,42,48], and sleep problems (present in 29%) [35,41,44,46,48] were the most prevalent, followed by symptoms of distress, extraversion, and neuroticism (each present in 6% of these articles) [30,33]. Similarly, Clark et al. [31] found evidence that, among the most frequent personality types in patients with CPMP, defensive–very anxious stood out. 

The analysis of existing correlations among symptoms of depression and the other symptoms studied in six of these articles (35%) [30,33,35,43,44,48] showed positive correlations with anxiety, neuroticism, somatic symptoms and their severity, the improper use of opioids, avoidance behaviour, coping strategies, and pain management; negative correlations were found with extraversion, the acceptance of chronic pain, beliefs concerning pain management, and happiness. Complementary to this, Neumann and Hampel [40] pointed out that the greater the chronic nature of the pain, the greater the severity of the depressive symptoms.

The analysis of existing correlations among symptoms of anxiety and the other symptoms studied in five of these articles (29%) [30,31,33,35,48] showed positive correlations with depression, distress, neuroticism, high sensorial sensitivity, the improper use of opioids, avoidance behaviour, coping strategies, and pain management; negative correlations were found with the acceptance of chronic pain, beliefs concerning pain management. and happiness. With respect to the consumption of opioids, no differences were noted concerning the presence of symptoms of depression or anxiety in patients with CPMP.

With respect to sleep problems in CPMP, the most prevalent dysfunction was insomnia. Pakpour et al. [41] and Skarpsno et al. [46] showed evidence that sleep problems were related to a worse recuperation and an increase in the intensity of pain. In accordance with the logistic regression model of Ho et al. [35], patients with insomnia showed almost twice the probability of suffering CPMP as compared with those who did not suffer insomnia.

It is worth mentioning the study of Kasahara et al. [36], which evaluated the presence of symptoms pertinent to attention deficit/hyperactivity disorder (ADHD) in patients with CPMP. The results showed significantly higher positive scores than in the general population in Inattention/Memory Problems, Hyperactivity/Restlessness, Problems with Self-Concept, DSM-IV Inattentive Symptoms, DSM-IV Hyperactive–Impulsive Symptoms, DSM-IV ADHD Symptoms Total, and ADHD Index. Furthermore, the dimensions of Hyperactivity/Restlessness and Hyperactive–Impulsive Symptoms correlated positively with the intensity of pain. On the other hand, Fujii et al. [34] evaluated the quality of life for patients with CPMP. They found significant associations between a greater load of somatic symptoms and a lower quality of life with respect to health independent of depression and the number of comorbid illnesses.

#### 3.5.3. Neuropsychological Dysfunctions

The neuropsychological dysfunctions of CPMP and its subtypes were evaluated in a total of seven articles (35%) [29,32,37,38,42,45,47]. The cognitive functions evaluated in two of these articles (29%) showed heterogeneous results. In the study by Brown et al. [29], patients with CPMP showed no significant differences in attention or working memory compared with participants with no pathology. However, in the study by Richards et al. [42], patients with CPMP did indeed show significantly lower performance in attention and working memory compared with participants with no pathology. It should be pointed out that no differences were found in the consumption of opioids when evaluating cognitive functions in the group with CPMP.

Resting-state functional connectivity (rsFC) was evaluated in two of these articles (29%). Coppieters et al. [32] and Shen et al. [45] showed evidence of significant dysfunction in patients with CPMP. To be precise, in the study by Coppieters et al. [32], they presented a higher performance in rsFC in the amygdala (associated with the processing and regulation of affection and the processing of possible threats) and the left frontal operculum (the region in the ventrolateral prefrontal cortex associated with sensory discrimination of the pain process and the cognitive–affective implications of pain) compared with participants with no pathology. Similarly, in the study by Shen et al. [45], patients with CPMP obtained a higher performance score on the rsFC of the visual networks (in charge of guiding the visual attention resources), thus allowing a fairly accurate differentiation (79.3%) between patients with CPMP and those with no such diagnosis. 

In the study by Kim et al. [37], patients with CPLBP showed greater connectivity of the salience network with the pons, the cerebellum, and the primary somatosensory cortex (S1) to the nociceptive stimuli applied to the lower back compared with participants with no pathology. The increase in the intensity of low back pain following pain exacerbation manoeuvres was related to a greater connectivity between the primary somatosensory cortex and the anterior insula. The salience network is associated with the reallocation of attention resources towards outstanding stimuli, such as pain, and is strongly influenced by pain catastrophizing. 

In the study by Ma et al. [38], patients with CPMP presented significantly lower levels of cognitive empathy and emotional disengagement with respect to participants with no pathology. The results of the fMRI showed evidence of multiple abnormal connections related to the anterior insula, the left parietal lobe, and the left dorsolateral cortex. The analysis of the correlations among these variables came close to statistical significance, which suggested that CPLBP was the principal cause of the reduction in the attention resources towards external stimuli because of their reallocation towards internal regulation. This was why they obtained low scores in emotional disengagement. Furthermore, according to the study by Tabira et al. [47], for patients with CPLBP at risk of becoming chronic, symptoms of depression and pain catastrophizing presented an attention bias towards threatening stimuli and negative emotional expressions.

## 4. Discussion

The last revision of the ICD-11 [9] established a new definition for chronic pain with multifactorial origins. The revision defined chronic primary musculoskeletal pain (CPMP) as being located in muscles, bones, joints, or tendons and associated with significant emotional distress and/or functional disability. This new conceptualisation supposes an opportunity to obtain a better understanding of this diagnosis when the aetiology of the illness is not clear.

The systematic review carried out in this paper responds to the above conceptualisation of the ICD-11 by analysing the psychopathological symptomatology and the neuropsychological dysfunctions of CPMP. This review was carried out on 20 articles [29,30,31,32,33,34,35,36,37,38,39,40,41,42,43,44,45,46,47,48], providing a reliable, up-to-date summary of the psychological consequences of CPMP.

In general, most of the studies provide evidence concerning the severity of chronic pain in the evaluated participants, pointing towards levels of pain between moderate and severe. All the evidence indicates that as the intensity of pain increases, so does the participants’ psychopathological symptomatology (depression, anxiety, insomnia, lack of attention, and hyperactivity/impulsiveness), as well as the use of maladaptive coping strategies [33,36,43,44]. Similarly, the levels of pain intensity also seem to be related to a lower self-confidence in being able to manage the symptoms related to pain and a lower acceptance of this pathology [33,36,43,44]. Another manifestation present in most patients with CPMP is pain catastrophizing, which shows similar relationships to those of pain intensity with the abovementioned symptomatology [33]. It should be pointed out that in 18 of the reviewed articles [30,31,33,34,35,36,37,38,39,40,41,42,43,44,45,46,47,48], the subtype of CPMP is chronic primary low back pain (CPLBP). Only two of the studies (17%), i.e., Brown et al. [29] and Coppieters et al. [32], analyse the chronic primary cervical pain (CPCP) subtype.

As for psychopathological symptomatology, the analysis of the reviewed studies shows coherent and homogeneous results. Depression, anxiety, and insomnia are the most prevalent symptoms in most participants. In accordance with the results observed, the symptoms of depression and anxiety are related to an increase in the participants’ psychopathological symptomatology and the use of maladaptive coping strategies. Furthermore, high levels of symptoms of depression and anxiety are also associated with a decrease in self-confidence to manage the pain-related symptoms, a lower willingness to accept the diagnosis of CPMP, and a decrease in the levels of happiness [30,31,33,34,43,44].

Similarly, the studies that evaluate the presence of sleep problems and their consequences in patients with CPMP coincide in the fact that the development or persistence of this type of problem is related to a deficient recuperation and a greater intensity of pain [35,41,44,46,48]. In the study by Ho et al. [35], the participants with insomnia present almost double the probability of suffering from CPMP as compared with those without insomnia. Thus, sleep problems are a relevant risk factor in the development of this pathology. With respect to other risk factors, the studies by Ho et al. [35], Neumann and Hampel [40], and Weiner et al. [48] also point out that depression, anxiety, coping strategies, age, and the level of physical activity are all associated with the presence of CPMP.

Regarding neuropsychological dysfunctions, the reviewed studies show evidence of dysfunctions in the cerebral structures related to attention and the processing of pain in patients with CPMP, including the somatosensory cortex, the insular cortex, the anterior cingulate cortex, the amygdala, and the prefrontal cortices [32,37,38,45]. In addition, the studies by Coppieters et al. [32] and Shen et al. [45] also show evidence of an increase in resting-state functional connectivity (rsFC) in patients with CPMP in comparison with those with no pathology. These dysfunctions have repercussions for the functioning of the brain and seem to be related to an attention bias towards threatening stimuli, stimuli related to pain, and negative emotional expressions, as well as an increase in internal emotional self-regulation. This also implies a reduction in the capacity to concentrate on the rest of the external stimuli, as well as a greater disengagement with respect to the emotional state of others. The demonstrated deficits in the domain of attention are coherent with the results of the study by Kasahara et al. [36], which show evidence of the presence of symptoms pertaining to attention deficit and hyperactivity disorder (ADHD) in patients with CPMP, such as a lack of attention and hyperactivity/impulsiveness.

On the other hand, the results obtained in the study by Brown et al. [29] are also worth noting. Of the reviewed articles, this is the only one in which patients with CPMP present low levels of pain intensity and disability associated with pain. Furthermore, in the neuropsychological evaluation, there are no significant differences with respect to the participants without any pathology. However, in those studies in which the intensity of pain and the disability associated with that pain are high, the results of the neuropsychological evaluation show differences with respect to the general population [32,37,42]. These results allow us to suppose that what has been set out above in psychopathological symptomatology is accurate in the sense that the greater the intensity of pain and the disability it causes, the greater the dysfunctions will be on a neuropsychological level. The findings from the rest of the studies that evaluate the cognitive functions of patients with CPMP, especially in executive functions, are not enough to determine the existence of deficits in these domains [29,42].

Because of the wide variety of symptoms in the diagnosis of chronic pain, treatment cannot be limited to interventions from a single discipline [49,50]. Recent studies indicate that this condition requires an integrated combination of different perspectives, thus necessitating an interdisciplinary approach [51,52]. To achieve optimal treatment, interventions must combine pharmacological and non-pharmacological treatments, including psychological interventions, physiotherapy, and physical activity programmes [51,52,53]. Nevertheless, the current classification of chronic pain by the WHO highlighted the need to identify effective treatments within the new diagnostic categories [54,55].

Although this research focused on identifying the psychological profile of patients with CPMP, the adoption of methodologies based on systematic reviews also has great potential in the treatment of chronic pain. In this sense, conducting systematic reviews with criteria aligned with those established in the ICD-11 can help identify the most effective interventions for pain reduction, both from a clinical and academic perspective. Additionally, the standardisation of treatments and the improvement of clinical guidelines in the management of CPMP could benefit both patients and healthcare professionals, providing an evidence-based guide for clinical practice.

Apart from the results set out above, it is important to point out the existence of limitations in the studies that make up the present systematic review, which must be taken into account when interpreting the results.

First, the incorporation of new diagnostic categories in the ICD-11 for chronic pain implies the existence of contradictions with some already published articles. Of these contradictions, the most important one that affects the present review to a great extent is the distinction between CPMP and fibromyalgia, which are now understood as different conditions. According to the WHO [9], the main difference between both diagnoses lies in the fact that fibromyalgia is a chronic pain that affects at least four of the five regions of the body, while CPMP is limited to muscles, bones, joints, or tendons. Nevertheless, many of the reviewed studies did not take this distinction into account, understanding fibromyalgia to be a condition linked to CPMP [56,57,58,59,60]. Because of this situation, the total number of articles included in this review is notably reduced.

Similarly, also related to the new diagnostic categories of the ICD-11, the subtypes of CPMP represent another limitation. The predominant subtype in most of the reviewed articles is chronic primary low back pain, followed by chronic primary cervical pain; however, none of the participants had a diagnosis of primary thoracic pain or chronic primary pain in the limbs. This situation poses a problem when generalising the results demonstrated in the current work.

Second, there are some limitations related to the samples. Although many of the studies included a control group with no pathologies for comparing results [29,32,35,37,38,39,42,44,45], eleven only had a group with CPMP [30,31,33,34,36,40,41,43,46,47,48]. In addition, the sample size of one of the studies was below twenty subjects [47]. As for the ages of the participants, eight of the studies did not establish a maximum age for participation [29,33,36,38,41,42,46,47].

Third, the instruments used to evaluate the participants represent another limitation. The instruments aimed at evaluating psychopathological symptomatology are very heterogeneous despite the fact that the symptoms and/or dysfunctions they measure are generally the same. Furthermore, the majority of these tests are self-reporting; this means that the demonstrated results may be influenced by variables not controlled by the researchers, such as social desirability or a lack of understanding and mood, among others. Regarding neuropsychological dysfunctions, it should be pointed out that most of the studies were not designed with the aim of evaluating these aspects. Except for the studies by Brown et al. [29] and Richards et al. [42], the rest did not use neuropsychological evaluation tests to measure cognitive functioning; they only used MRIs or EEGs [32,37,38,45,47]. Although the results of these tests can be used to check whether there are dysfunctions in the structure and functioning of the brain, they cannot determine the severity of the deficits in cognitive domains with total accuracy [61].

In summary, the findings presented suggest that as the intensity of pain increases, so does the psychopathological symptomatology. However, the number of studies examining this relationship is limited. Furthermore, we can verify that depression, anxiety, and sleep problems are usually common symptoms of CPMP. Similarly, the relationship between CPMP and deficits on a cognitive level is complex. Although dysfunctions in the attention domain seem to be common, there is no sufficient evidence to determine the existence of deficits in the rest of the cognitive functions.

## 5. Conclusions

The present systematic review examines the works published over the last five years concerning the psychopathological symptoms and the dysfunctions that exist on a neuropsychological level in chronic primary musculoskeletal pain. The new conceptualisation of CPMP, and the distinction with respect to other diagnoses that have been complementary until recently, supposes an advance towards understanding it as an independent entity. As pointed out by Meints and Edwards [3] and Moreira et al. [39] in their studies, this review suggests that the presence of chronic pain, concretely, CPMP, not only involves a series of consequences on the cognitive, emotional, and behavioural levels but that the cerebral functions also worsen. Despite the limitations that were identified, we consider that the results shown in this present review contribute to an improved understanding of the psychopathological symptomatology and neuropsychological dysfunctions of this pathology. In addition, the results have great relevance for the development and optimisation of multidisciplinary treatments adapted to the deficits produced as a consequence of this diagnosis. Therefore, identifying the needs of the population with CPMP is crucial for interventions to achieve success, as has been achieved in this research.

## Figures and Tables

**Figure 1 healthcare-12-01465-f001:**
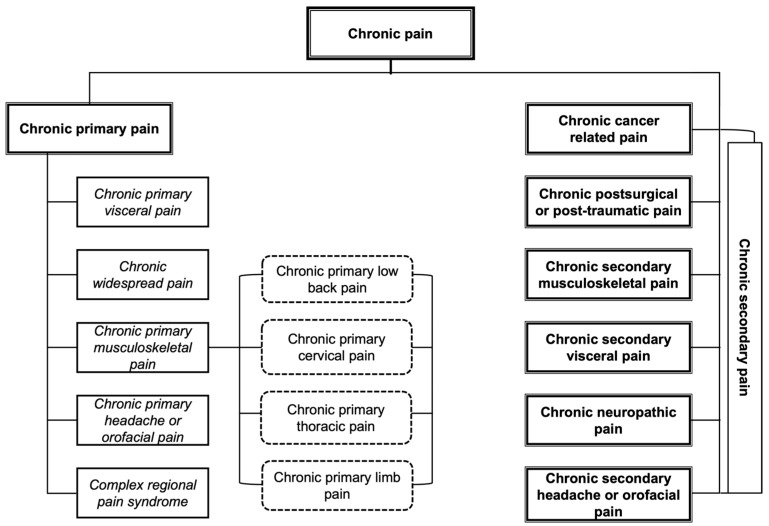
Classification of chronic pain according to the ICD-11. Adapted from Treede et al. [10].

**Figure 2 healthcare-12-01465-f002:**
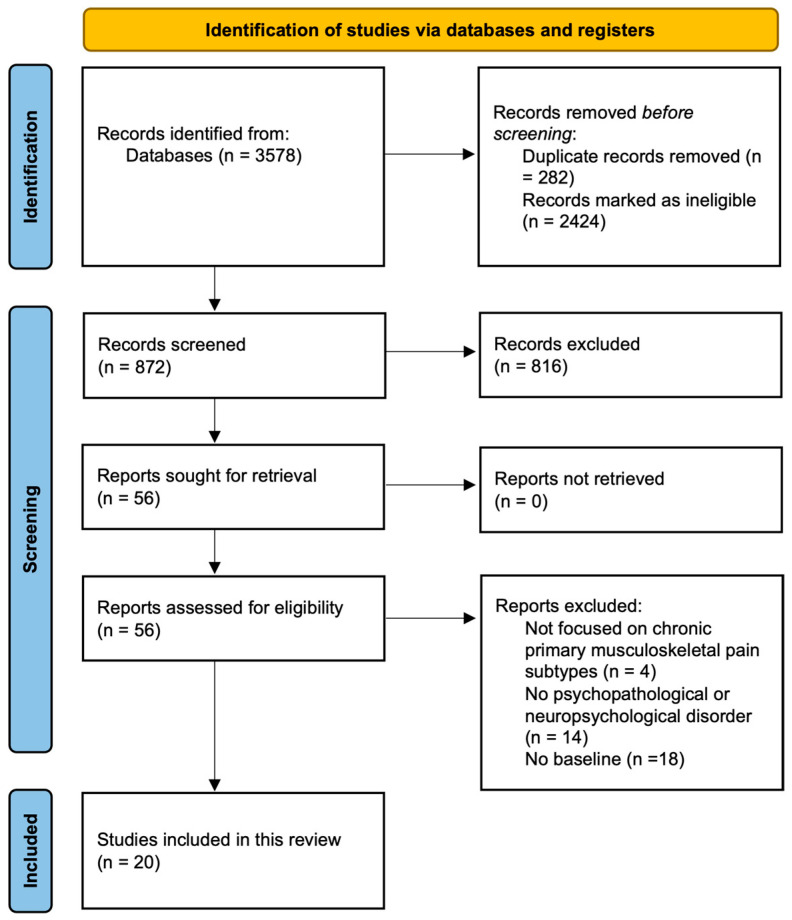
PRISMA flow diagram. Adapted from Page et al. [27].

**Table 1 healthcare-12-01465-t001:** Characteristics of the studies included in this review.

	Author	Publication Year	Country	Design	Subtypes
1.	Brown et al. [29]	2022	The USA	Cross-sectional study	Chronic primary cervical pain (CPCP)
2.	Brunner et al. [30]	2018	Germany	Cross-sectional study	Chronic primary low back pain (CPLBP)
3.	Clark et al. [31]	2019	New Zealand	Cross-sectional study	Chronic primary low back pain (CPLBP)
4.	Coppieters et al. [32]	2021	Belgium	Cross-sectional study	Chronic primary cervical pain (CPCP)
5.	Day et al. [33]	2019	Australia	Cross-sectional study	Chronic primary low back pain (CPLBP)
6.	Fujii et al. [34]	2018	Japan	Cross-sectional study	Chronic primary low back pain (CPLBP)
7.	Ho et al. [35]	2019	Norway	Cross-sectional study	Chronic primary low back pain (CPLBP)
8.	Kasahara et al. [36]	2021	Japan	Cross-sectional study	Chronic primary low back pain (CPLBP)
9.	Kim et al. [37]	2019	The USA	Cross-sectional study	Chronic primary low back pain (CPLBP)
10.	Ma et al. [38]	2020	China	Cross-sectional study	Chronic primary low back pain (CPLBP)
11.	Moreira et al. [39]	2021	Brazil	Cross-sectional study	Chronic primary low back pain (CPLBP)
12.	Neumann and Hampel [40]	2022	Germany	Longitudinal study	Chronic primary low back pain (CPLBP)
13.	Pakpour et al. [41]	2018	Iran	Longitudinal study	Chronic primary low back pain (CPLBP)
14.	Richards et al. [42]	2018	Australia	Cross-sectional study	Chronic primary low back pain (CPLBP)
15.	Rogers et al. [43]	2022	The USA	Cross-sectional study	Chronic primary low back pain (CPLBP)
16.	Rumble et al. [44]	2021	The USA	Longitudinal study	Chronic primary low back pain (CPLBP)
17.	Shen et al. [45]	2019	The USA	Cross-sectional study	Chronic primary low back pain (CPLBP)
18.	Skarpsno et al. [46]	2020	Norway	Longitudinal study	Chronic primary low back pain (CPLBP)
19.	Tabira et al. [47]	2020	Japan	Cross-sectional study	Chronic primary low back pain (CPLBP)
20.	Weiner et al. [48]	2019	The USA	Cross-sectional study	Chronic primary low back pain (CPLBP)

**Table 2 healthcare-12-01465-t002:** Psychopathological dysfunctions.

Authors	Sample	Gender and Age	Instruments	Results
Brunner et al. [30]	*n* = 49	Men = 24 (49%) Women = 25 (51%) Average age = 47.08	NRS RMDQ SBT 4DSQ TSK	The sample with CPLBP showed the following correlations: - The scores of the SBT scale presented moderately positive correlations with the scores of the TSK scale (r = 0.59; *p* < 0.05). - The scores of the SBT scale presented moderately positive correlations with the scores of the 4DSQ scale, concretely in the dimensions of anxiety (r = 0.50; *p* < 0.05), depression (r = 0.43; *p* < 0.05), and distress (r = 0.63; *p* < 0.05). - The scores of the TSK scale presented regular positive correlations with the scores of the 4DSQ scale, concretely in the dimensions of anxiety (r = 0.13; *p* < 0.05), depression (r = 0.33; *p* < 0.05), and distress (r = 0.25; *p* < 0.05). Within the 4DSQ scale there existed moderately positive correlations between the scores of distress and depression (r = 0.56; *p* < 0.05), distress and anxiety (r = 0.66; *p* < 0.05), and depression and anxiety (r = 0.43; *p* < 0.05).
Clark et al. [31]	*n* = 165	Men = 39 (23.6%) Women = 126 (76.4%) Average age = 45	CSI AASP STAI MCSDS	The sample with CPLBP presented different sensory profiles as a result of combining the neurological thresholds to sensory stimuli with the adaptive behavioural response to sensory stimuli. The sensory profiles presented in the sample were one or more of the following: high trait sensory sensitivity (low threshold, passive response; 55%, *n* = 91), sensation avoidance (low threshold, active response; 44%, *n* = 72), low registration (high threshold, passive response; 36%, *n* = 60), and low trait sensation-seeking (high threshold, active response; 38%, *n* = 62). The proportions of the types of personality in the sample were defensive high anxious (45%; *n* = 75), high anxious (25%; *n* = 43), and repressor (25%; *n* = 41). The sample with CPLBP presented the following correlations: - The scores of the CSI presented moderate positive correlations with the sensory profiles, concretely with high trait sensory sensitivity (r = 0.63: *p* < 0.01), sensation avoidance (r = 0.48; *p* < 0.01), and a low trait sensation seeking (r = 0.54; *p* < 0.01). - The scores of the anxiety trait presented moderate positive correlations with the profile of high trait sensory sensitivity (r = 0.43; *p* < 0.01). As for the other profiles, the correlation was weak. - The scores of the anxiety trait also presented moderate positive correlations with the scores of the CSI (r = 0.46; *p* < 0.01). The linear regression model showed evidence that the increase in the scores of the CSI could be predicted through sensory hypersensitivity profiles with a low register, the scores of the anxiety trait, and the highly defensive and very anxious personality type.
Day et al. [33]	*n* = 69	Men = 33 (48%) Women = 36 (52%) Average age = 51	NRS PCS SOPA PROMIS SHS NRP–Avoidance scale CPAQ-8 BFI EEG	The sample with CPLBP showed the following correlations: - The intensity of the pain presented regular positive correlations with pain catastrophizing (r = 0.33; *p* < 0.01), depression (r = 0.24; *p* < 0.05), and anxiety (r = 0.23; NS), while it presented moderate negative correlations with the beliefs concerning pain control (r = −0.41; *p* < 0.01) and regular negative correlations with the acceptance of chronic pain (r = −0.33; *p* < 0.01). - Pain catastrophizing presented moderate positive correlations with depression (r = 0.44; *p* < 0.01), anxiety (r = 0.45; *p* < 0.01), avoidance behaviour (r = 0.63; *p* < 0.01), and neuroticism (r = 0.49; *p* < 0.01), while it presented moderate negative correlations with beliefs concerning pain control (r = −0.50; *p* < 0.01) and the acceptance of chronic pain (r = −0.49; *p* < 0.01) and regular negative correlations with happiness (r = −0.34; *p* < 0.01). - Depression presented moderate positive correlations with anxiety (r = 0.67; *p* < 0.01), avoidance behaviour (r = 0.40; *p* < 0.01), and neuroticism (r = 0.66; *p* < 0.01), while it presented moderate negative correlations with happiness (r = −0.63; *p* < 0.01) and the acceptance of chronic pain (r = −0.58; *p* < 0.01) and regular negative correlations with the beliefs concerning pain control (r = −0.34; *p* < 0.01) and extraversion (r = −0.30; *p* < 0.01). - Anxiety presented moderate positive correlations with avoidance behaviour (r = 0.41; *p* < 0.01) and neuroticism (r = 0.61; *p* < 0.01), while it presents moderate negative correlations with the acceptance of chronic pain (r = −0.48; *p* < 0.01) and regular negative correlations with beliefs concerning pain control (r = −0.23; NS) and happiness (r = −0.39; *p* < 0.01).
Fujii et al. [34]	*n* = 3100	Men = 1617 (52.2%) Women = 1483 (47.8%) Average age = 44.5	SSS-8 PHQ-2 EQ-5D	The sample with CPLBP showed the following correlations: - The total score of the SSS-8 scale presented moderately negative correlations with the scores of the EQ-5D scale (r = −0.55; *p* < 0.0001). As the score of the PHQ-2 scale increased, so did the proportion of the sample with very high scores of the SSS-8 scale; in addition, the score of the PHQ-2 scale was also associated with that obtained using the EQ-5D scale. In accordance with the regression model, a greater load of somatic symptoms in the sample with CPLBP was significantly associated with a lower quality of life with respect to health, independently of depression and the number of co-morbid illnesses (high score as opposed to very high score on SSS-8: β = 0.040; *p* < 0.0001/minimum score as opposed to very high score on SSS-8: β = 0.22; *p* < 0.0001).
Ho et al. [35]	*n* = 30,669 - CPLBP (6559). - Control (24,140).	Men = 14,006 (46%) Women = 16,663 (54%) Average age = 52.2	Clinical diagnoses from the HUNT Study 3 (2006–2008)	In the entire sample, 6.1% (*n* = 1871) of the participants presented insomnia and 2.4% insomnia and CPLBP (n = 719). The logistic regression model showed evidence that insomnia (OR = 2.46; *p* < 0.001), age (OR = 1.01; *p* < 0.001), physical activity (OR = 0.88; *p* < 0.001), depression (borderline: OR = 1.66; *p* < 0.001/possible: OR = 2.40; *p* < 0.001), and anxiety (borderline: OR = 1.73; *p* < 0.001/possible: OR = 2.48; *p* < 0.001) were all significantly associated with the presence of CPLBP. In the multiple logistic regression model, which included the said factors, those with insomnia presented almost double the probabilities of suffering from CPLBP as compared with those who did not suffer from insomnia (OR = 1.99; *p* < 0.0001).
Kasahara et al. [36]	*n* = 60	Men = 29 (48.3%) Women = 31 (51.7%) Average age = 54.9	CAARS CAARS-S CAARS–O NRS	Within the sample suffering from CPLBP, 48.3% (*n* = 29) obtained positive scores on CAARS-S and 60% (*n* = 36) obtained positive scores on CAARS-O. Overall, 76.6% (*n* = 46) obtained positive scores on CAARS-S or CAARS-O, and 31.1% (*n* = 19) obtained positive scores on both. The results obtained were compared with those obtained by the general population in another study in which the same instruments were administered. On both the CAARS-S and the CAARS-O scales, the group with CPLBP had significantly higher scores on the subscales of Inattention/Memory Problems (*p* < 0.05; *p* < 0.05), Hyperactivity/Restlessness (*p* < 0.001; *p* < 0.05), Problems with Self-Concept (*p* < 0.05; *p* < 0.001), DSM-IV Inattentive Symptoms (*p* < 0.005; *p* < 0.05), DSM-IV Hyperactive–Impulsive Symptoms (*p* < 0.005; NS), DSM-IV ADHD Symptoms Total (*p* < 0.001; NS), and the ADHD Index (*p* < 0.001; *p* < 0.005) with respect to the general population. Similarly, evidence of the following correlations was found: - The subscale of Hyperactivity/Restlessness on the CAARS-S presented a regular positive correlation with the maximum intensity of pain (r = 0.27; *p* < 0.05). - The subscale of Hyperactive–Impulsive Symptoms on the CAAR-S presented a regular positive correlation with the duration of the pain (r = 0.34; *p* < 0.01). - The subscale of Hyperactivity/Restlessness on the CAARS-O presented a moderate positive correlation with the maximum intensity of pain (r = 0.44; *p* < 0.001) and with the average intensity of pain (r = 0.44; *p* < 0.001). - The subscale of DSM-IV Hyperactive–Impulsive Symptoms of the DSM-IV on the CAARS-O presented a regular positive correlation with the average intensity of pain (r = 0.28; *p* < 0.05).
Moreira et al. [39]	*n* = 36 - CPLBP (18). - Control (18).	Men = 10 (26.6%) Women = 28 (77.7%) Average age = 24.2	CPM Protocol NRS SF-MPQ TSK BDI VAS PCS RMDQ	The group with CPLBP presented levels of intensity of pain (*p* < 0.01), symptoms of depression (*p* < 0.05), and symptoms of anxiety (*p* < 0.01) that were significantly higher than those of the control group. Furthermore, they present significantly higher levels of pain and mood symptoms (*p* < 0.01; *p* < 0.01), fear of movement and physical activity (*p* < 0.05), and pain catastrophizing (*p* < 0.05) with respect to the control group. The group with CPLBP showed a significantly lower pain pressure threshold (13.5%) with respect to the control group, as well as a considerable exacerbation of cognitive–behavioural changes.
Neumann and Hampel. [40]	*n* = 526	Men = 94 (17.9%) Women = 432 (82.1%) Average age = 53.2	MPSS CES-D HADS Mini-SCL SF-12 DSF	The sample with CPLBP was classified into three progressive states of pain depending on the pain’s level of chronification as follows: state of pain I (*n* = 126), state of pain II (*n* = 270), and state of pain III (*n* = 130). Among these three states there existed statistically significant differences in the duration of pain (*p* < 0.05), the location of pain (*p* < 0.01), and the average intensity of pain (*p* < 0.01); the differences were greater as the chronification of pain advanced. Similarly, there were statistically significant differences in the presence of depressive symptoms among the different groups (*p* < 0.01); the differences were greater as the chronification of pain advanced. As for the state of health, there were statistically significant differences for physical health (*p* < 0.01) but not for mental health.
Pakpour et al. [41]	*n* = 761	Men = 414 (55.4%) Women = 347 (44,6%) Average age = 41.15	VAS PSQI HADS	At the start of the study, 48% (*n* = 365) of the sample with CPLBP stated that they had problems sleeping in the previous month, increasing to a total of 67.6% (*n* = 514) after six months of monitoring. Regarding the intensity of pain, 38.3% of the sample presented severe levels of pain after six months of monitoring. The logistical regression model presented the following results: - Those with problems sleeping at the start of the study had a 50% possibility of having a deficient recuperation (OR = 1.50; *p* < 0.05) and more than double the possibility of having pain of greater intensity (OR = 2.48; *p* < 0.05) after six months of monitoring. - Those who did not manifest problems sleeping at the start of the study, but who developed this problem later, had almost double the possibility of having a deficient recuperation (OR = 2.17; *p* < 0.05) and almost triple the possibility of having a greater intensity of pain (OR = 2.88; *p* < 0.05) after six months of monitoring. - Those with persistent problems sleeping, both at the start of the study and during monitoring, had almost triple the possibility of having a deficient recuperation (OR = 2.95; *p* < 0.05) and more than triple the possibility of having a greater intensity of pain (OR = 3.45; *p* < 0.05) after six months of monitoring. Those with problems sleeping at the start of the study, but not during monitoring, had reduced probability of not recuperating (OR = 0.50; *p* < 0.05) and of having a greater intensity of pain (OR = 0.49; *p* < 0.05) after six months of monitoring.
Rogers et al. [43]	*n* = 294	Men = 92 (31.1%) Women = 202 (68.9%) Average age = 45.8	BPI ODSIS SSASI OPMM COMM	The sample with CPLBP declared having suffered, on average, for 4.30 years, with an average intensity of 6.54/10. Regarding the state of abusive consumption of opioids, 56.7% (*n* = 167) classified themselves as abusive consumers. In addition, the following correlations were found: - The intensity of pain presented regular positive correlations with depressive symptoms (r = 0.26; *p* < 0.01), the symptoms of anxiety sensitivity (r = 0.20; *p* < 0.01), coping motives (r = 0.13; *p* < 0.05), and pain management (r = 0.14; *p* < 0.05). - Depressive symptoms presented a moderate positive correlation with the symptoms of anxiety sensitivity (r = 0.55; *p* < 0.01) and the abusive use of opioids (r = 0.45; *p* < 0.01) and a regular positive correlation with coping motives (r = 0.37; *p* < 0.01) and pain management (r = 0.35; *p* < 0.01). - The symptoms of anxiety sensitivity presented a moderate positive correlation with coping motives (r = 0.47; *p* < 0.01), pain management (r = 0.48; *p* < 0.01), and the abusive use of opioids (r = 0.42; *p* < 0.01). - The coping motives presented a moderate positive correlation with pain management (r = 0.71; *p* < 0.01) and the abusive use of opioids (r = 0.51; *p* < 0.01); furthermore, pain management presented a moderate positive correlation with the abusive use of opioids (r = 0.51; *p* < 0.01). In accordance with the hierarchical regression model, anxiety sensitivity was significantly associated with coping motives (R2 = 0.29; *p* < 0.01) and coping with pain (R2 = 0.27; *p* < 0.01). In addition, anxiety sensitivity was indirectly associated with the state of the abusive use of opioids through both motives (OR = 1.03; *p* < 0.05/OR = 1.09; *p* < 0.05).
Rumble et al. [44]	*n* = 117 - CPLBP (82). - Control (35).	Men = 54 (46.2%) Women = 63 (53.8%) Average age = 43.4	GCPS CES-D The STOP-BANG Home sleep monitoring through the Actiwatch2 Sleep diaries	The group with CPLBP presented levels of pain intensity (*p* < 0.01), depressive symptoms (*p* < 0.01), wake after sleep onset (*p* < 0.01), and time spent in bed (*p* < 0.05) significantly higher than those of the control group. In addition, the group with CPLBP presented quality of sleep (*p* < 0.01) and of refreshed sleep (*p* < 0.01) significantly below that of the control group. - The intensity of pain presented regular positive correlations with sleep latency (r = 22; *p* < 0.05), while it presented regular negative correlations with sleep quality (r = −0.28; *p* < 0.05) and refreshed sleep (r = −0.31; *p* < 0.01). - Depressive symptoms presented regular positive correlations with the intensity of CPLBP (r = 36; *p* < 0.01) and with wake after sleep onset (r = 0.27; *p* < 0.01), while they presented moderate negative correlations with quality of sleep (r = −0.49; *p* < 0.01), refreshed sleep (r = −0.50; *p* < 0.01), and pain status (r = −0.54; *p* < 0.01). All the sleep variables were significantly correlated with each other (*p* < 0.05).
Skarpsno et al. [46]	*n* = 6200	Men = 2488 (40%) Women = 3712 (60%) Average age = 49.7	Information obtained from the HUNT Study 2 (1995–1997) and the HUNT Study 3 (2006–2008)	Within the sample suffering from CPLBP, the women (RR = 0.65) and men (RR = 0.81) who frequently/always experienced insomnia had lower probabilities of recuperating from CPLBP as compared with those who did not suffer from insomnia. The probability of recuperating from CPLBP was inversely associated with the number of symptoms of insomnia in women (one symptom: RR = 0.81; two: RR = 0.68; three: RR = 0.60) and, to a lesser extent, in men (one symptom: RR = 0.99; two: RR = 0.84; three: RR = 0.82). Both women and men with CPLBP had a lower probability of recuperation (RR = 0.46; 0.67) if they stated that they always/often suffered from insomnia in comparison with those who rarely/never suffered from insomnia (RR = 0.68; RR = 0.81)
Weiner et al. [48]	*n* = 47	Men = 41 (87.2%) Women = 6 (12.8%) Average age = 68	NRS RMDQ PHQ-9 GAD-7 The Insomnia Severity Index CSQ FABQ	Within the sample suffering from CPLBP, 83% (*n* = 39) presented factors that contribute to pain in the central nervous system. Concretely, 38.3% (*n* = 18) presented moderate levels of anxiety and depression, 63.8% (*n* = 30) moderate levels of insomnia, and 63.8% (*n* = 30) a catastrophizing cognition with respect to CPLBP and a maladaptive coping style with respect to physical activity. The presence of anxiety and depression (*p* < 0.05), insomnia (*p* < 0.01), and maladaptive coping strategies (*p* < 0.05) were significantly associated with more intense levels of pain.

Abbreviations: CPLBP = chronic primary low back pain; NS = not significant; NRS = the Numeric Rating Scale; RMDQ = the Roland–Morris Disability Questionnaire; SBT = STarT back screening tool; 4DSQ = the Four-Dimensional Symptom Questionnaire; TSK = the Tampa Scale of Kinesiophobia; CSI = the Central Sensitization Inventory; AASP = Adolescent/Adult Sensory Profile Questionnaire; STAI = the State–Trait Anxiety Inventory; MCSDS = the Marlowe Crowne Social Desirability Scale; PCS = the Pain Catastrophizing Scale; SOPA = the Survey of Pain Attitudes; PROMIS = the Patient-Reported Outcomes Measurement Information System; SHS = the Subjective Happiness Scale; NRP = the Negative Responsivity to Pain Avoidance scale; CPAQ-8 = the Chronic Pain Acceptance Questionnaire; BFI = the Big Five Inventory; EEG = continuous electroencephalogram; SSS-8 = the Japanese version of the Somatic Symptom Scale-8; PHQ-2 = Patient Health Questionnaire-2 Depression Scale; EQ-5D = The European Quality of Life-5; HUNT = The Nord–Trøndelag Health Study; CAARS = the Conners’ Adult ADHD Rating Scales; CAARS-S = the Conners’ Adult ADHD Rating Scales self-report; CAARS–O = the Conners’ Adult ADHD Rating Scales–Observer; CPM Protocol = Conditioned Pain Modulation Protocol; SF-MPQ = Short-form McGill Pain Questionnaire; BDI = Beck Depression Inventory; VAS = Visual Analogue Scale; MPSS = the Mainz Pain Staging System; CES-D = The Center for Epidemiological Studies—Depression; HADS = Hospital Anxiety and Depression Scale; Mini-SCL = the German short version of the Brief-Symptom-Checklist; SF-12 = 12-Item Short Form Health Survey; DSF = the German Questionnaire of Pain; PSQI = the Pittsburgh Sleep Quality Index; BPI = the Brief Pain Inventory; ODSIS = the Overall Depression Symptom and Impairment Scale; SSASI = the Short Scale Anxiety Sensitivity Index; OPMM = the Opioid prescription medication motives; COMM = the current opioid misuse measure; GCPS = the Graded Chronic Pain Scale; PHQ-9 = Patient Health Questionnaire Depression Scale; GAD-7 = Generalized Anxiety Disorder Scale; CSQ = the coping strategy questionnaire; FABQ = the Fear-Avoidance Beliefs Questionnaire.

**Table 3 healthcare-12-01465-t003:** Neuropsychological dysfunctions.

Authors	Sample	Gender and Age	Instruments	Results
Brown et al. [29]	*n* = 38 - CPCP (17). - Control (21).	Men =13 (34.2%) Women = 25 (65.8%) Average age = 24.6	Demographic questionnaire NDI NRS Digital cognitive assessments developed by Cogstate: attention/reaction time, verbal working memory, and working memory	The group with CPCP presented a light level of pain intensity, as well as low levels of disability associated with neck pain. The group with CPCP did not present statistically significant differences in the tests of verbal working memory: duration (ms) (*p* = 0.726), verbal working memory: correct responses (*p* = 0.417), attention: speed (ms) (*p* = 0.426), attention: errors (*p* = 0.974), working memory: speed (ms) (*p* = 0.771), or in working memory: errors (*p* = 0.424) with respect to the control group.
Coppieters et al. [32]	*n* = 107 - CPCP (38). - Neck pain—origin, physical trauma (37). - Control (32).	Women = 107 (100%) Average age = 32.6	VNRS-11 NDI CSI QST Digital pressure algometer fMRI	The group with CPCP and the group with neck pain as a consequence of a physical trauma presented statistically significant results in the intensity of neck pain (*p* < 0.001), disability related to neck pain (*p* < 0.001), and in the scores of the CSI, with respect to the control group. The results of the fMRI showed that the group with CPCP and the group with trauma pain presented better performance in resting state functional connectivity (rsFC) between the left amygdala (associated with the processing and regulation of affection and the processing of possible threats) and the left frontal operculum (the region in the ventrolateral frontal cortex associated with sensory discrimination of the processing of pain and the cognitive–affective implications of pain) in comparison with the control group (*p* < 0.001). The results were associated with a decrease in the endogenous inhibition of pain in the groups with CPCP and trauma pain and with a greater number of symptoms with self-reported central sensitivity in the CPCP group (*p* = 0.02). These associations implied a link between cognitive–affective and sensory modulations in CPCP.
Kim et al. [37]	*n* = 181 - CPLBP (127). - Control (54).	Men = 80 (44.2%) Women = 101 (55.8%) Average age = 39.4	BDI BPSD PROMIS PCS fMRI	The group with CPLBP presented significantly higher levels of depressive symptoms (*p* < 0.01), intensity of pain (*p* < 0.01), and pain catastrophizing (*p* < 0.01) with respect to the control group. The group with CPLBP presented a greater connectivity of the salience network with the pons, the cerebellum, and the primary somatosensory cortex (S1) to the nociceptive stimuli applied to the lower back with respect to the control group. In pain exacerbation manoeuvres, the group with CPLBP presented greater posterior connectivity of the primary somatosensory cortex with different cerebral regions of the salience network including the following: the anterior insular cortex, dorsolateral prefrontal cortex, and anterior temporoparietal junction. The increase in the intensity of low back pain following pain exacerbation manoeuvres presented a regular positive correlation with a greater connectivity between the somatosensory cortex (S1) and the left anterior insula (r = 0.36; *p* < 0.05). The salience network was closely linked to the ventral attention network and associated with the reallocation of attention resources towards outstanding stimuli, such as pain. Furthermore, the said connectivity was found to be strongly influenced by pain catastrophizing,
Ma et al. [38]	*n* = 46 - CPLBP (24). - Control (22).	Men =17 (37%) Women = 29 (63%) Average age = 32	VAS BES-A: - Emotional contagion. - Emotional disconnection. - Cognitive empathy. fMRI	The group with CPLBP presented significantly lower levels in cognitive empathy (*p* = 0.0015), emotional disengagement (*p* = 0.0017), and total scores on the BES-A scale (*p* = 0.005) with respect to the control group; however, there were no significant differences in emotional contagion (*p* = 0.119). The results of the fMRI in the group with CPLBP showed evidence of multiple abnormal pathways in the brain centred on the anterior insula. Within the group with CPLBP, the abnormal functional connection state of the left parietal lobe and the left dorsolateral prefrontal cortex led, as a result, to an incorrect allocation of attention resources. Although there was no existing correlation between the connectivity of either of these and the scores in emotional disengagement, the results were close to being statistically significant (r = 0.39; *p* = 0.058). The said results suggested that CPLBP was the main cause of the reduction in the attention resources towards external stimuli because of the reallocation towards internal self-regulation, this being the reason why they obtain low scores in emotional disengagement (self-protection regulation mechanism).
Richards et al. [42]	*n* = 60 - CPLBP-OP (18). - CPLBP-NO (22). - Control (20).	Men = 34 (56.7%) Women = 26 (43.3%) Average age = 60.6	BPI DASS-21 PSEQ PCS Battery of cognitive tests: - The Wechsler Test of Adult Reading. - The California Verbal Learning Test. - The Everyday Memory Questionnaire—Revised. - The Brief Assessment of Prospective Memory. - The California Older Adult Stroop Test. - Letter–Number Sequencing. - Matrix Reasoning.	The groups with CPLBP with opiate consumption (OP) and without opiate consumption (NO) presented significantly higher levels of depressive symptoms (*p* < 0.001/*p* < 0.001), anxiety (*p* < 0.001/*p* < 0.001), and stress (*p* < 0.001/*p* < 0.001) with respect to the control group; however, there were no significant differences between the groups in these dimensions. Nor were there significant differences between the groups in the average scores of the intensity of chronic pain, the interference of pain in the activities of daily life, or pain catastrophizing. Regarding cognitive functions, the groups with CPLBP with and without consumption of opioids presented a significantly lower performance in memory (*p* < 0.01/*p* < 0.01) and attention (*p* < 0.01/*p* < 0.05) with respect to the control group. Regarding executive functions, the groups with CPLBP with and without the consumption of opioids presented significantly lower performance with respect to the control group, concretely in working memory (*p* < 0.05/*p* < 0.05). There were no significant differences between the groups in these dimensions.
Shen et al. [45]	*n* = 164 - CPLBP (90). - Control (74).	Men = 69 (42%) Women = 95 (58%) Average age = 33.4	Pain Bothersomeness Scale BDI-II fMRI	The group with CPLBP presented very low scores in depression. The analysis of the resting state functional connectivity (rsFC) showed evidence that when the group with CPLBP used the primary visual network (in charge of orienting the visual attention resources), there were significant increases in rsFC in the right postcentral (S1) and precentral (M1) gyri and decreases in rsFC in the left angular gyrus/lateral occipital cortex. The results showed a significant alteration in the rsFC of the visual networks in the group with CPLBP, which obtained better performance than the control group. The rsFC between the primary visual network and the primary somatosensory cortex (S1: a critical component of the nociceptive pathway) presented a regular negative correlation with the duration of the CPLBP (r = −0.24; *p* < 0.05). In addition, the rsFC of the visual network allowed the participants with CPLBP and those of the control group to be differentiated with an accuracy of 79.3% (*p* < 0.001).
Tabira et al. [47]	*n* = 13	Men = 2 (15.4%) Women = 11 (84.6%) Average age = 70.3	EEG Japanese version of: - JLEQ. - FABQ. - J-SBST. - PCS. - BDI-II.	The sample with CPLBP presented the following correlations: - The scores obtained on the J-SBST presented high negative correlations with the amplitude N1 of Cz (r = −0.65; *p* < 0.05). The amplitude N1 was associated with capturing visual sensory attention based on the prominence of the stimulus and could be attributed to a greater effort to redirect attention to the visual threat stimuli. - The scores obtained in the BDI-II presented high negative correlations with the amplitude of P2 in Cz (r = −0.74; *p* < 0.05) and in Pz (r = −0.60; *p* < 0.05). The amplitude P2 was associated with the processing of emotions on faces and was a sensitive neural response to the stimuli related with threats. - The scores obtained in the PCS presented high negative correlations with the amplitude of P2 in Fz (r = −0.63; *p* < 0.05), in Cz (r = −0.70; *p* < 0.05), and in Pz (r = −0.61; *p* < 0.05), which was associated with the processing of emotions on faces. Having CPLBP and high scores on J-SBST, BDI-II, and PCS implied a positive attention bias towards threatening stimuli and negative emotional expressions.

Abbreviations: CPCP = chronic primary cervical pain; CPLBP = chronic primary low back pain; MS = milliseconds; NDI = the Neck Disability Index; NRS = the Numeric Rating Scale; rsFC = resting-state functional connectivity; VNRS-11 = the 11-point verbal numeric rating scale; CSI = Central Sensitization Inventory; QST = quantitative sensory testing; fMRI = functional magnetic resonance imaging; BDI = Beck Depression Inventory; BPSD = Back Pain-Specific Disability; PROMIS = the Patient-Reported Outcomes Measurement Information System; PCS = the Pain Catastrophizing Scale; VAS = Visual Analogue Scale; BES-A: the Basic Empathy Scale in Adults; BPI = the Brief Pain Inventory; DASS-21 = the Depression Anxiety Stress Scales; PSEQ = the Pain Self-Efficacy Questionnaire; BDI-II = the Beck Depression Inventory-II; EEG = electroencephalogram; JLEQ = Japan Low Back Pain Evaluation Questionnaire; FABQ = the Fear-Avoidance Beliefs Questionnaire; J-SBST = STarT back screening tool.

## Data Availability

No new data were created or analysed in this study. Data sharing is not applicable to this article.

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
