# Peer review of "Systematic Review of the Psychopathological Symptomatology and Neuropsychological Disorders of Chronic Primary Musculoskeletal Pain"

_healthcare, 2024, doi:10.3390/healthcare12151465_

Round 1

Reviewer 1 Report

Comments and Suggestions for Authors

In this manuscript, Arevalo-Martinez et al. summarized the psychopathological symptomatology and neuropsychological disorders of CPMP from 20 articles and compares them to the new conceptualisation in the ICD-11. This manuscript provided a clear description of the methodology and a fairly comprehensive and logical analysis. Authors found a lack of researches on the CPMP, but on the other hand, the number of articles used for analysis in this manuscript seemed insufficient, especially the data testing conditions for these analyzed articles were uneven. Generally, the topic is interesting, the content is well organized, and the conclusion is appropriate.  

Minor comments:

1. Page 10 and 16, Table 2 and 3: The captions of the two tables are incorrect.

2. Page 21, lines 387-390: please add references.

3. Title: Authors would use "CPMP" in the title instead of "chronic pain".

Author Response

3. Point-by-point response to Comments and Suggestions for Authors

In this manuscript, Arevalo-Martinez et al. summarized the psychopathological symptomatology and neuropsychological disorders of CPMP from 20 articles and compares them to the new conceptualisation in the ICD-11. This manuscript provided a clear description of the methodology and a fairly comprehensive and logical analysis. Authors found a lack of researches on the CPMP, but on the other hand, the number of articles used for analysis in this manuscript seemed insufficient, especially the data testing conditions for these analyzed articles were uneven. Generally, the topic is interesting, the content is well organized, and the conclusion is appropriate.  

Comments 1: Page 10 and 16, Table 2 and 3: The captions of the two tables are incorrect.

Response 1: Thank you for pointing this out. We agree with this comment. Therefore, we have corrected the captions of Tables 2 and 3. The updated captions can now be found on pages 10 and 16.

Comments 2: Page 21, lines 387-390: please add references.

Response 2: The authors agree with the reviewer's comment. We have, accordingly, added the necessary references to emphasize this point. Due to the revisions suggested by the reviewers, the page numbering has changed compared to the previous version of the manuscript.

Original text: page 21, lines 387-390.

Updated text: page 19, lines 422-425.

The references [33,36,43,44] have been added on page 19, lines 422-425. These references were included in the original manuscript and are as follows:

33.     Day, M.A.; Matthews, N.; Newman, A.; Mattingley, J.B.; Jensen, M.P. An Evaluation of the Behavioral Inhibition and Behavioral Activation System (BIS-BAS) Model of Pain. Rehabil. Psychol. 2019, 64, 279–287, doi:10.1037/rep0000274.

36.     Kasahara, S.; Niwa, S.-I.; Matsudaira, K.; Sato, N.; Oka, H.; Fujii, T.; Konno, S.-I.; Kikuchi, S.-I.; Yamada, Y. High Attention-Deficit/Hyperactivity Disorder Scale Scores Among Patients with Persistent Chronic Nonspecific Low Back Pain. Pain Physician 2021, 24, E299–E307.

43.     Rogers, A.H.; Garey, L.; Raines, A.M.; Allan, N.P.; Schmidt, N.B.; Zvolensky, M.J. Anxiety Sensitivity and Opioid Use Motives among Adults with Chronic Low Back Pain. Exp. Clin. Psychopharmacol. 2022, 30, 23–30, doi:10.1037/pha0000381.

44.     Rumble, D.D.; O’Neal, K.; Overstreet, D.S.; Penn, T.M.; Jackson, P.; Aroke, E.N.; Sims, A.M.; King, A.L.; Hasan, F.N.; Quinn, T.L.; et al. Sleep and Neighborhood Socioeconomic Status: A Micro Longitudinal Study of Chronic Low-Back Pain and Pain-Free Individuals. J. Behav. Med. 2021, 44, 811–821, doi:10.1007/s10865-021-00234-w.

Text where citations have been added: In general, most of the studies provide evidence concerning the severity of chronic pain in the evaluated participants, pointing towards levels of pain between moderate and severe. All the evidence indicates that, as the intensity of the pain increases, so does the participants’ psychopathological symptomatology (depression, anxiety, insomnia, lack of attention and hyperactivity/impulsiveness), as well as the use of maladaptive coping strategies [33,36,43,44].

Comments 3: Title: Authors would use "CPMP" in the title instead of "chronic pain".

Response 3: Thank you for the suggestion. We have made the change in the title as requested.

Original title: Systematic review of the psychopathological symptomatology and neuropsychological disorders of chronic pain

Modified title: Systematic review of the psychopathological symptomatology and neuropsychological disorders of chronic primary musculoskeletal pain.

Reviewer 2 Report

Comments and Suggestions for Authors

The background and objective of this study are as follows.

---->"There are several intervening biological, psychological, and social factors involved in its appearance that significantly affect the activities of daily life. It is also associated with significant emotional anxiety and/or functional disability. This review systematically analyses works published in the last five years that evaluate the psychopathological symptomatology and the neuropsychological disorders of chronic primary musculoskeletal pain (CPMP)." from the Abstract of this article------

However, upon examining the analysis content, there is no systematic review of how therapeutic interventions have been conducted in various fields for chronic pain. It is necessary to reexamine the study's objective and review the description in the body of the paper. The purpose of conducting a systematic review in a clinical setting must be clear in order to establish clear criteria for selecting papers to analyze and to ensure that the analysis is focused on deriving clinically necessary content. It is essential to clearly redefine the study's objective and appropriately revise the criteria for selecting papers and the analysis content.

Comments on the Quality of English Language

Rather than simply describing the analysis content, it is necessary to succinctly explain the core content in line with the analytical objectives. Repeated abbreviations in the table should be uniformly indicated below the table. The table design should also be standardized according to the design recommended by the journal, and the analysis content should be concisely described.

Author Response

Point-by-point response to Comments and Suggestions for Authors

Comments 1: The background and objective of this study are as follows. ---->"There are several intervening biological, psychological, and social factors involved in its appearance that significantly affect the activities of daily life. It is also associated with significant emotional anxiety and/or functional disability. This review systematically analyses works published in the last five years that evaluate the psychopathological symptomatology and the neuropsychological disorders of chronic primary musculoskeletal pain (CPMP)." from the Abstract of this article------

However, upon examining the analysis content, there is no systematic review of how therapeutic interventions have been conducted in various fields for chronic pain. It is necessary to reexamine the study's objective and review the description in the body of the paper. The purpose of conducting a systematic review in a clinical setting must be clear in order to establish clear criteria for selecting papers to analyze and to ensure that the analysis is focused on deriving clinically necessary content. It is essential to clearly redefine the study's objective and appropriately revise the criteria for selecting papers and the analysis content.

Response 1: Thank you for your comments. While we understand the reviewer's rationale for making this comment, the authors believe that implementing this modification would change the purpose of this research.

As outlined throughout the introduction, CPMP represents a new diagnostic category in the latest revision of the ICD-11 by the World Health Organization. In this context, the definition of precise diagnostic criteria and the establishment of a new hierarchical classification serve as a starting point for better understanding this diagnosis and its associated consequences.

Therefore, we consider that before addressing therapeutic interventions, it is necessary to identify the psychopathological symptomatology and neuropsychological disorders associated with CPMP. In this way, the findings provided by this research will constitute a solid foundation for future investigations and the development of effective therapeutic interventions.

Hence, the main objective of this research is to systematically identify and analyse the psychopathological symptomatology and neuropsychological disorders associated with CPMP. However, to justify focusing on the psychological profile of this population at baseline rather than on therapeutic interventions, we have added the following text on page 3, lines 116-120.

Added text: In line with the conclusions raised by Treede et al. [10], we are currently at the right moment to enhance the understanding of chronic primary pain and develop multimodal treatments tailored to the new diagnostic classifications. Therefore, to intervene comprehensively in the population with CPMP, it is essential to thoroughly assess their needs beforehand.

However, if the editor considers it appropriate, changes will be made.

 Response to Comments on the Quality of English Language

Comments 2: Rather than simply describing the analysis content, it is necessary to succinctly explain the core content in line with the analytical objectives.

Response 2: The main purpose of this research has been to construct a theoretical framework on CPMP, focusing on the psychopathological and neuropsychological profile associated with this condition. Due to the methodological diversity of the studies included in the review, we chose to begin with a detailed description of the content of each study. This approach has allowed us to conduct a comprehensive analysis that highlights common aspects in the psychopathological symptomatology and neuropsychological disorders in individuals with CPMP, as outlined in the Discussion section, pages 19-21. We consider this initial description essential to facilitate a subsequent interpretation of the results of each study.

The ultimate aim of this research is to use the findings to design therapeutic intervention programmes that improve the psychological adjustment of these patients. However, to justify this approach, we have added the following text in the Conclusion, page 21, lines 541-543.

Added text: Therefore, identifying the needs of the population with CPMP is crucial for interventions to achieve success, as has been done in this research.

However, if the editor considers it appropriate, changes will be made.

Comments 3: Repeated abbreviations in the table should be uniformly indicated below the table.

Response 3: Agree. We have, accordingly, added the abbreviations below Tables 2 and 3. Specifically, abbreviations were added below Table 2 on page 15, lines 384-398, and below Table 3 on page 18, lines 400-407.

Added abbreviations below Table 2: CPLBP = Chronic primary low back pain; NS = Not significant; NRS = The Numeric Rating Scale; RMDQ = The Roland-Morris Disability Questionnaire; SBT = STarT back screening tool; 4DSQ = The Four-Dimensional Symptom Questionnaire; TSK = The Tampa Scale of Kinesiophobia; CSI = The Central Sensitization Inventory; AASP = Adolescent/Adult Sensory Profile Questionnaire; STAI = The State-Trait Anxiety Inventory; MCSDS = The Marlowe Crowne Social Desirability Scale; PCS = The Pain Catastrophizing Scale; SOPA = The Survey of Pain Attitudes; PROMIS = The Patient-Reported Outcomes Measurement Information System; SHS = The Subjective Happiness Scale; NRP = The Negative Responsivity to Pain–Avoidance scale; CPAQ-8 = The Chronic Pain Acceptance Questionnaire; BFI = The Big Five Inventory; EEG = Continuous electroencephalogram; SSS-8 = The Japanese version of the Somatic Symptom Scale-8; PHQ-2 = Patient Health Questionnaire-2 Depression Scale; EQ-5D = The European Quality of Life-5; The HUNT = The Nord-Trøndelag Health Study; CAARS = The Conners' Adult ADHD Rating Scales; CAARS-S = The Conners' Adult ADHD Rating Scales self-report; CAARS–O = The Conners' Adult ADHD Rating Scales–Observer; CPM Protocol = Conditioned Pain Modulation Protocol; SF-MPQ = Short-form McGill Pain Questionnaire; BDI = Beck Depression Inventory; VAS = Visual Analog Scale; MPSS = The Mainz Pain Staging System; CES-D = The Center for Epidemiological Studies-Depression; HADS = Hospital Anxiety and Depression Scale; Mini-SCL = The German short version of the Brief-Symptom-Checklist; SF-12 = 12-Item Short Form Health Survey; DSF = The German Questionnaire of Pain; PSQI = The Pittsburgh Sleep Quality Index; BPI = The Brief Pain Inventory; ODSIS = The Overall Depression Symptom and Impairment Scale; SSASI = The Short Scale Anxiety Sensitivity Index; OPMM = The Opioid prescription medication motives; COMM = The Current opioid misuse measure; GCPS = The Graded Chronic Pain Scale; PHQ-9 = Patient Health Questionnaire Depression Scale; GAD-7 = Generalized Anxiety Disorder Scale; CSQ = The coping strategy questionnaire; FABQ = The Fear-Avoidance Beliefs Questionnaire.

Added abbreviations below Table 3: CPCP = Chronic primary cervical pain; CPLBP = Chronic primary low back pain; MS = Milliseconds; NDI = The Neck Disability Index; NRS = The Numeric Rating Scale; rsFC = resting state functional connectivity; VNRS-11 = The 11-point verbal numeric rating scale; CSI = Central Sensitization Inventory; QST = Quantitative sensory testing; fMRI = Functional magnetic resonance imaging; BDI = Beck Depression Inventory; BPSD = Back-Pain Specific Disability; PROMIS = The Patient-Reported Outcomes Measurement Information System; PCS = The Pain Catastrophizing Scale; VAS = Visual Analog Scale; BES-A: The Basic Empathy Scale in Adults; BPI = The Brief Pain Inventory; DASS-21 = The Depression Anxiety Stress Scales; PSEQ = The Pain Self-Efficacy Questionnaire; BDI-II = The Beck Depression Inventory-II; EEG = Electroencephalogram; JLEQ = Japan Low Back Pain Evaluation Questionnaire; FABQ = The Fear Avoidance-Beliefs Questionnaire; J-SBST = STarT back screening tool.

Comments 4: The table design should also be standardized according to the design recommended by the journal, and the analysis content should be concisely described.

Response 4: Thank you for pointing this out. We agree with this comment. We have updated the headers of Tables 1, 2, and 3 to ensure that the legend for each column remains always visible. We have also adjusted the formatting of the text and the tables to prevent the rows corresponding to each study from being cut off.

Furthermore, all the information analysed in the tables has been used to identify the psychological and/or neuropsychological assessment tests frequently used in patients with CPMP and to review the results of empirical studies that evaluate psychopathological symptomatology and neuropsychological disorders in this population. However, to be more concise and facilitate the analysis of the information presented, we have simplified Tables 2 and 3 by using only abbreviations in the instrument column and have added the abbreviations at the end of the table as suggested by the reviewer in the previous comment.

Reviewer 3 Report

Comments and Suggestions for Authors

Dear authors,

Congratulations on your excellent work on the systematic review of psychopathological symptomatology and neuropsychological disorders of chronic pain. Your comprehensive approach and adherence to the Preferred Reporting Items for Systematic Reviews and Meta-Analyses (PRISMA) guidelines resulted in a comprehensive and well-structured review. The meticulous attention to detail in the selection criteria, search strategy and quality assessment demonstrates a deep commitment to providing valuable information on the psychopathological and neuropsychological aspects of CPMP. Although the basis of the review is sound, there are some minor aspects that, if integrated, could further enhance the clarity and comprehensiveness of the study.

INTRODUCTION

- I suggest expanding the literature review to include a broader range of studies, ensuring a more comprehensive understanding of the current state of CPMP research.

-Although the focus on CPMP is justified by its prevalence, the introduction does not sufficiently explain why this specific type of chronic pain is chosen over others.I recommend providing a more detailed justification for the focus on CPMP, possibly discussing its unique features or gaps in current research that this review aims to fill.

Materials and Methods

-  although the inclusion and exclusion criteria are listed, "the sample must not present additional diseases" is a vague criterion, make explicit which ones

Author Response

Point-by-point response to Comments and Suggestions for Authors

Dear authors,

Congratulations on your excellent work on the systematic review of psychopathological symptomatology and neuropsychological disorders of chronic pain. Your comprehensive approach and adherence to the Preferred Reporting Items for Systematic Reviews and Meta-Analyses (PRISMA) guidelines resulted in a comprehensive and well-structured review. The meticulous attention to detail in the selection criteria, search strategy and quality assessment demonstrates a deep commitment to providing valuable information on the psychopathological and neuropsychological aspects of CPMP. Although the basis of the review is sound, there are some minor aspects that, if integrated, could further enhance the clarity and comprehensiveness of the study.

Introduction

Comments 1: I suggest expanding the literature review to include a broader range of studies, ensuring a more comprehensive understanding of the current state of CPMP research.

Response 1: Thank you for pointing this out. We found the reviewer's comment very useful. To ensure a more comprehensive understanding of CPMP, we have expanded the literature reviewed and added the following text on page 3, lines 97-110.

Original text: Fitzcharles et al. [11] have proposed a series of differentiating characteristics for chronic primary musculoskeletal pain as opposed to chronic secondary musculoskeletal pain. Of these, it is worth noting a greater prevalence of psychopathologies, cognitive deterioration and other co-morbid conditions. They also point to a closer relation between pain and psychosocial stress, as well as to deterioration in the quality of life equal to, or even greater than, that of patients with CPMP.

Added text: Although research aiming to deepen the understanding of CPMP is limited, Fitzcharles et al. [19] have proposed several distinctive features of CPMP compared to chronic secondary musculoskeletal pain. In this regard, CPMP is predominantly nociceptive and described as a sharp, stabbing, or burning sensation [8,21]. Individuals with this diagnosis often exhibit pain hypersensitivity, as well as a variable and diffuse pain experience that may radiate to other areas of the body [22,23]. Moreover, increased pain intensity is common and correlates with experienced psychosocial stress [4,10]. CPMP has also been associated with higher prevalence of psychopathologies, cognitive impairment, and other comorbid conditions [4,19]. On the other hand, chronic secondary musculoskeletal pain is predominantly nociceptive [10,18]. This type of pain is described as a localized pressure sensation near the injury, with infrequent hypersensitivity and stable pain intensity [24,25]. Although quality of life is generally impaired in both diagnoses, CPMP may lead to greater deterioration due to its high prevalence of psychopathological and physical comorbidities [4,18,26].

In addition, we have added the corresponding references:

18.     El-Tallawy, S.N.; Nalamasu, R.; Salem, G.I.; LeQuang, J.A.K.; Pergolizzi, J.V.; Christo, P.J. Management of Musculoskeletal Pain: An Update with Emphasis on Chronic Musculoskeletal Pain. Pain Ther 2021, 10, 181–209, doi:10.1007/s40122-021-00235-2.

21.     Bułdyś, K.; Górnicki, T.; Kałka, D.; Szuster, E.; Biernikiewicz, M.; Markuszewski, L.; Sobieszczańska, M. What Do We Know about Nociplastic Pain? Healthcare 2023, 11, 1794, doi:10.3390/healthcare11121794.

22.     Clauw, D.J.; Essex, M.N.; Pitman, V.; Jones, K.D. Reframing Chronic Pain as a Disease, Not a Symptom: Rationale and Implications for Pain Management. Postgraduate Medicine 2019, 131, 185–198, doi:10.1080/00325481.2019.1574403.

23.   Fitzcharles, M.-A.; Cohen, S.P.; Clauw, D.J.; Littlejohn, G.; Usui, C.; Häuser, W. Nociplastic Pain: Towards an Understanding of Prevalent Pain Conditions. The Lancet 2021, 397, 2098–2110, doi:10.1016/S0140-6736(21)00392-5.

24.     Abd-Elsayed, A.; Deer, T.R. Different Types of Pain. In Pain: A Review Guide; Abd-Elsayed, A., Ed.; Springer International Publishing: Cham, 2019; pp. 15–16 ISBN 978-3-319-99124-5.

25.   Prescott, S.A.; Ratté, S. Chapter 23 - Somatosensation and Pain. In Conn’s Translational Neuroscience; Conn, P.M., Ed.; Academic Press: San Diego, 2017; pp. 517–539 ISBN 978-0-12-802381-5.

26.     Zhuang, J.; Mei, H.; Fang, F.; Ma, X. What Is New in Classification, Diagnosis and Management of Chronic Musculoskeletal Pain: A Narrative Review. Front Pain Res (Lausanne) 2022, 3, 937004, doi:10.3389/fpain.2022.937004.

Comments 2: Although the focus on CPMP is justified by its prevalence, the introduction does not sufficiently explain why this specific type of chronic pain is chosen over others.I recommend providing a more detailed justification for the focus on CPMP, possibly discussing its unique features or gaps in current research that this review aims to fill.

Response 2: Agree. We have, accordingly, developed the justification for focusing on this specific type of chronic pain. Although the primary reason for focusing on CPMP is its high prevalence compared to other types of chronic pain, the significant impact it has on the physical, psychological, and social well-being of those affected is also an additional reason. Therefore, we have added the following text on pages 2-3, lines 76-85.

Added text: Epidemiological studies conducted globally have shown a significant increase in diagnoses of chronic pain in recent decades [11,12,13]. Specifically, the most frequently reported types of chronic pain are musculoskeletal in origin, often localized in areas such as the back, shoulders, neck, hips, knees, wrists, and feet [1,14,15]. Additionally, alongside physical and psychological distress, individuals with chronic musculoskeletal pain also experience difficulties in engaging in social activities and performing their work tasks effectively [16,17].

Due to its high prevalence compared to other chronic pain conditions and the significant impact the disease has on the lives of those who experience it, this systematic review will focus on chronic primary musculoskeletal pain (CPMP) [1,18].

In addition, we have added the corresponding references:

11.       Cheung, C.W.; Choi, S.W.; Wong, S.S.C.; Lee, Y.; Irwin, M.G. Changes in Prevalence, Outcomes, and Help-Seeking Behavior of Chronic Pain in an Aging Population Over the Last Decade. Pain Practice 2017, 17, 643–654, doi:10.1111/papr.12496.

12.       Hecke, O. van; Torrance, N.; Smith, B.H. Chronic Pain Epidemiology and Its Clinical Relevance. British Journal of Anaesthesia 2013, 111, 13–18, doi:10.1093/bja/aet123.

13.       Zajacova, A.; Grol-Prokopczyk, H.; Zimmer, Z. Pain Trends Among American Adults, 2002–2018: Patterns, Disparities, and Correlates. Demography 2021, 58, 711–738, doi:10.1215/00703370-8977691.

14.       Breivik, H.; Eisenberg, E.; O’Brien, T. The Individual and Societal Burden of Chronic Pain in Europe: The Case for Strategic Prioritisation and Action to Improve Knowledge and Availability of Appropriate Care. BMC Public Health 2013, 13, 1-14, doi:10.1186/1471-2458-13-1229.

15.       Yong, R.J.; Mullins, P.M.; Bhattacharyya, N. Prevalence of Chronic Pain among Adults in the United States. PAIN 2022, 163, e328-e332, doi:10.1097/j.pain.0000000000002291.

16.    Bonanni, R.; Cariati, I.; Tancredi, V.; Iundusi, R.; Gasbarra, E.; Tarantino, U. Chronic Pain in Musculoskeletal Diseases: Do You Know Your Enemy? Journal of Clinical Medicine 2022, 11, 1-27, doi:10.3390/jcm11092609.

17.       Demircioğlu, A.; Özkal, Ö.; Dağ, O. Multiple Factors Affecting Health-Related Quality of Life in Women With Chronic Multisite Musculoskeletal Pain: A Cross-Sectional Study in Ankara, Turkey. Eval Health Prof 2022, 45, 115–125, doi:10.1177/01632787211049273.

18.       El-Tallawy, S.N.; Nalamasu, R.; Salem, G.I.; LeQuang, J.A.K.; Pergolizzi, J.V.; Christo, P.J. Management of Musculoskeletal Pain: An Update with Emphasis on Chronic Musculoskeletal Pain. Pain Ther 2021, 10, 181–209, doi:10.1007/s40122-021-00235-2.

Materials and Methods

Comments 3: Although the inclusion and exclusion criteria are listed, "the sample must not present additional diseases" is a vague criterion, make explicit which ones

Response 3: Thank you for the suggestion. We have made the change in the inclusion criteria. This change can be found on page 4, lines 133-134.

Original text: the sample should not present additional illnesses.

Added text: the sample should not have additional diseases, such as cardiovascular, respiratory, metabolic, or neurological conditions.

Round 2

Reviewer 2 Report

Comments and Suggestions for Authors

Thank you for promptly incorporating reviewers' feedback into the article. Through this paper, I hope to provide valuable insights and theoretical background to researchers in the field. In my personal opinion, it would be beneficial to add a discussion on how the analysis method of systematic reviews based on research topics can be applied in the medical field. Particularly, suggesting how such analysis can be utilized in treating pain reduction methods with clinical and academic proposals would be great. To facilitate immediate understanding in clinical settings, it would be helpful to convey the key points concisely, as some sentences may be overly verbose. Thank you to all researchers for their hard work.

Author Response

3. Point-by-point response to Comments and Suggestions for Authors

Comments 1: In my personal opinion, it would be beneficial to add a discussion on how the analysis method of systematic reviews based on research topics can be applied in the medical field. Particularly, suggesting how such analysis can be utilized in treating pain reduction methods with clinical and academic proposals would be great. To facilitate immediate understanding in clinical settings, it would be helpful to convey the key points concisely, as some sentences may be overly verbose.

Response 1: Thank you for pointing this out. We agree with this comment. To ensure that we justify the potential of systematic reviews in identifying the most effective treatments from a clinical perspective in CPMP, we have added the following text to the manuscript on page 20, lines 482-498.

Added text: Due to the wide variety of symptoms in the diagnosis of chronic pain, treatment cannot be limited to interventions from a single discipline [49,50]. Recent studies indicate that this condition requires an integrated combination of different perspectives, thus necessitating an interdisciplinary approach [51,52]. To achieve optimal treatment, interventions must combine pharmacological and non-pharmacological treatments, including psychological interventions, physiotherapy, and physical activity programmes [51-53]. Nevertheless, the current classification of chronic pain by the WHO has highlighted the need to identify effective treatments within the new diagnostic categories [54,55].

Although this research has focused on identifying the psychological profile of patients with CPMP, the adoption of methodologies based on systematic reviews also has great potential in the treatment of chronic pain. In this sense, conducting systematic reviews with criteria aligned with those established in the ICD-11 can help identify the most effective interventions for pain reduction, both from a clinical and academic perspective. Additionally, the standardisation of treatments and the improvement of clinical guidelines in the management of CPMP could benefit both patients and healthcare professionals, providing an evidence-based guide for clinical practice.

In addition, we have added the corresponding references:

49.     Dydyk, A.M.; Conermann, T. Chronic Pain. In StatPearls; StatPearls Publishing: Treasure Island (FL), 2024.

50.     Gauthier, K.; Dulong, C.; Argáez, C. Multidisciplinary Treatment Programs for Patients with Chronic Non-Malignant Pain: A Review of Clinical Effectiveness, Cost-Effectiveness, and Guidelines – An Update; CADTH Rapid Response Reports; Canadian Agency for Drugs and Technologies in Health: Ottawa (ON), 2019; pp. 1-27.

51.     Scheidegger, A.; Gómez Penedo, J.M.; Blättler, L.T.; Aybek, S.; Bischoff, N.; grosse Holtforth, M. Improvements in Pain Coping Predict Treatment Success among Patients with Chronic Primary Pain. Journal of Psychosomatic Research 2023, 168, 111208, doi:10.1016/j.jpsychores.2023.111208.

52.     Hylands-White, N.; Duarte, R.V.; Raphael, J.H. An Overview of Treatment Approaches for Chronic Pain Management. Rheumatol Int 2017, 37, 29–42, doi:10.1007/s00296-016-3481-8.

53.     Semmons, J. The Role of Physiotherapy in the Management of Chronic Pain. Anaesthesia & Intensive Care Medicine 2019, 20, 440–442, doi:10.1016/j.mpaic.2019.05.012.

54.     Korwisi, B.; Barke, A.; Kharko, A.; Bruhin, C.; Locher, C.; Koechlin, H. Not Really Nice: A Commentary on the Recent Version of NICE Guidelines [NG193: Chronic Pain (Primary and Secondary) in over 16s: Assessment of All Chronic Pain and Management of Chronic Primary Pain] by the Pain Net. PAIN Reports 2021, 6, e961, doi:10.1097/PR9.0000000000000961.

55.     Smith, B.H.; Colvin, L.A.; Donaldson-Bruce, A.; Birt, A. Drugs for Chronic Pain: We Still Need Them. Br J Gen Pract 2021, 71, 172–172, doi:10.3399/bjgp21X715457.